# Pedestrian Trajectory Prediction with Missing Data: Datasets, Imputation, and Benchmarking

**Pranav Singh Chib**[1]    **Pravendra Singh**[1]

[1]Department of Computer Science and Engineering
Indian Institute of Technology
Roorkee, India
{pranavs_chib,pravendra.singh}@cs.iitr.ac.in

## Abstract

Pedestrian trajectory prediction is crucial for several applications such as robotics and self-driving vehicles. Significant progress has been made in the past decade thanks to the availability of pedestrian trajectory datasets, which enable trajectory prediction methods to learn from pedestrians' past movements and predict future trajectories. However, these datasets and methods typically assume that the observed trajectory sequence is complete, ignoring real-world issues such as sensor failure, occlusion, and limited fields of view that can result in missing values in observed trajectories. To address this challenge, we present TrajImpute, a pedestrian trajectory prediction dataset that simulates missing coordinates in the observed trajectory, enhancing real-world applicability. TrajImpute maintains a uniform distribution of missing data within the observed trajectories. In this work, we comprehensively examine several imputation methods to reconstruct the missing coordinates and benchmark them for imputing pedestrian trajectories. Furthermore, we provide a thorough analysis of recent trajectory prediction methods and evaluate the performance of these models on the imputed trajectories. Our experimental evaluation of the imputation and trajectory prediction methods offers several valuable insights. Our dataset provides a foundational resource for future research on imputation-aware pedestrian trajectory prediction, potentially accelerating the deployment of these methods in real-world applications. Publicly accessible links to the datasets and code files are available at https://github.com/Pranav-chib/TrajImpute.

## 1   Introduction

Pedestrian trajectory prediction [1, 2, 3, 4, 5, 6, 7] has various essential applications, such as self-driving automobiles, robot navigation, human behavior understanding, and more. These systems forecast the future trajectory of pedestrians based on their previously observed paths. Research in pedestrian trajectory prediction has significantly advanced in recent times due to the development of data-driven solutions and datasets. However, a predominant assumption in most current research is that the past observed coordinates of pedestrians are complete. This assumption does not hold in real-world scenarios [8] where sensor failures, limited field of view, and occlusion can lead to missing observations at any specific time instances, resulting in incomplete trajectories. This creates challenges for trajectory prediction tasks in real-world scenarios. To improve the effectiveness of trajectory prediction methods in real-world scenarios, they must anticipate and handle missing observed coordinates.

In multivariate time series, several imputation methods [9, 10, 11, 12] have emerged that address the issue of missing features by imputing them (filling of missing values). These methods [13, 14,

15, 16] utilize statistical and deep learning approaches and have achieved state-of-the-art results for imputation time series data. However, there has been limited exploration of imputation techniques in trajectory prediction [8, 17, 18], and there is a gap in the availability of imputation-centric pedestrian datasets, evaluation protocols, and benchmarks. We introduce TrajImpute, an imputation-centric trajectory prediction dataset, to address this. We have compiled commonly used pedestrian trajectory prediction datasets [19, 20], including ETH, HOTEL, UNIV, ZARA1, and ZARA2, (which are licensed for research purposes [1]) and introduced trajectories with missing observed coordinates. We follow two data generation strategies to simulate the missing coordinates: easy and hard modes. In the easy mode, we simulate scenarios where observed coordinates are missed for a shorter duration (could be continuous or discontinuous time frame). In contrast, the hard mode simulates scenarios where observed coordinates are missing for a longer duration. In addition to data generation, we benchmark several existing imputation methods [16, 14, 21, 9, 22, 23] on TrajImpute. We use these imputation methods to reconstruct the missing coordinates and evaluate their performance in both easy and hard modes. After extensive evaluation, we selected the best-performing imputation model and used its imputed data for the trajectory prediction task. The motivation of our work is to provide insights into how trajectory prediction models perform when missing coordinates are imputed. Additionally, we aim to understand how imputation methods perform on the pedestrian trajectory imputation task. Thus, TrajImpute provides a dataset in which missing coordinates are present in observed trajectories to simulate real-world scenarios and offers a unified framework for evaluating both imputation and trajectory prediction methods.

**Contributions.** We introduce TrajImpute, a trajectory prediction dataset designed to simulate missing coordinates in observed trajectories of pedestrians. TrajImpute bridges the gap between real-world scenarios and the rigid assumption that all coordinates are present in observed trajectories. We conduct extensive analyses and empirical studies to evaluate several existing imputation methods for the task of trajectory imputation on our TrajImpute dataset. Furthermore, we evaluate the performance of recent trajectory prediction methods on imputed data and provide insights for future development in this area. The dataset is provided under a Creative Commons CC BY-SA 4.0 license, allowing both academics and industry to use it.

## 2    Related Work

### 2.1    Imputation

RNN-based methods have initially been used for handling missing data in time series classification. Methods such as M-RNN [9] use the bidirectional RNN's hidden states for imputing missing values. BRITS [22] treats missing values as variables and considers feature correlations. Generative adversarial network approaches have also been applied. For instance, Luo et al. [24] propose GRU for imputation to capture temporal information in incomplete time series, serving as the basis for both the discriminator and generator in their GAN model. Additionally, Luo et al. [10] introduce $E^2GAN$, utilizing an auto-encoder to enhance imputation performance. Liu et al. propose NAOMI [10], a non-autoregressive model featuring a multiresolution decoder and bidirectional encoder, which was adversarially trained. Fortuin et al. [21] introduce GP-VAE, a variational auto-encoder method for time series imputation. It uses the Gaussian process (GP) prior in the latent space to represent the data. Furthermore, L-VAE [11] employs an additive multi-output GP-prior to accommodate additional covariate information alongside time for the imputation task. SGP-VAE [25] uses GP approximations to impute the missing values in spatiotemporal data. CNN-based methods such as TimesNet [16] use the fast Fourier transform to transform 1D time series into a 2D representation, making it easier to interpret data using CNNs. Recently, methods have started using the self-attention mechanism for data imputation. For instance, CDSA [13] uses cross-dimension attention for the imputation of missing data. DeepMVI [15] employs the transformer with convolutional window features and kernel regression. SAITS [14] uses joint training for both the imputation and reconstruction tasks to impute the missing values. SAITE employs two diagonally-masked self-attention mechanisms to capture the temporal dependency. However, self-attention-based studies for time-series imputation are still limited.

---

[1]See the statement at the top of https://icu.ee.ethz.ch/research/datsets.html and in the "Crowds Data" card of https://graphics.cs.ucy.ac.cy/portfolio.

Table 1: Comparison of different datasets for pedestrian trajectory prediction. The trajectory coordinates can be extracted from images (image-2D/3D) or directly from data containing only coordinates (world-2D/3D). Here, 2D/3D refers to the dimensionality of the coordinate output. The most widely used pedestrian trajectory prediction datasets are ETH, HOTEL, UNIV, ZARA1, and ZARA2. These reported datasets only contain complete coordinates and do not simulate missing coordinates.

| Datasets | Reference | Link | Availability | Citations | Coordinates | Pedestrian Data | Imputed Data |
|---|---|---|---|---|---|---|---|
| ETH | [20] | Link | ✓ | 1804 | world-2D | ✓ | × |
| HOTEL | [20] | Link | ✓ | | world-2D | ✓ | × |
| UNIV | [19] | Link | ✓ | 1247 | world-2D | ✓ | × |
| ZARA1 | [19] | Link | ✓ | | world-2D | ✓ | × |
| ZARA2 | [19] | Link | ✓ | | world-2D | ✓ | × |
| SDD | [51] | Link | ✓ | 872 | image-2D | ✓ | × |
| Trajnet | [52] | Link | ✓ | 244 | - | ✓ | × |
| GC | [53] | Link | ✓ | 279 | image-2D | ✓ | × |
| PETS | [54] | Link | ✓ | 711 | image-2D | ✓ | × |
| inD | [55] | Link | ✓ | 375 | world-2D | ✓ | × |
| Argoverse2 | [56] | Link | ✓ | 345 | world-3D | ✓ | × |
| KITTI | [57] | Link | ✓ | 144 | image-3D | ✓ | × |
| Ko-PER | [58] | Link | ✓ | 99 | world-2D | ✓ | × |
| TRAF | [59] | Link | ✓ | 280 | image-2D | ✓ | × |
| TrajImpute | (Our) | - | ✓ | - | world-2D | ✓ | ✓ |

## 2.2 Trajectory Prediction

Predicting an agent's future path based on their past observations is the objective of pedestrian trajectory prediction methods. Prior work on trajectory prediction [26, 26] involved using deterministic approaches, which use parameters such as acceleration and velocity to model pedestrian trajectories. With the advent of deep learning, researchers began sequence-to-sequence modeling of trajectories using Recurrent Neural Networks (RNNs) [27, 28] and Long Short-Term Memory networks (LSTM) [29] for sequence prediction of future trajectories. Some research generates multiple trajectory predictions using generative models like Variational Autoencoders (VAEs) [2, 30, 31, 32] and Generative Adversarial Networks (GANs) [33, 34, 35, 36], which consider the uncertainty of human trajectories. Diffusion models [37] are also being used to sample future trajectories but have high inference time due to costly denoising steps. Transformers [38, 39, 40, 41, 42, 43] can capture long-term temporal dependencies in pedestrian trajectories because of the self-attention mechanism. Additionally, to model social interactions between pedestrians and their surroundings, researchers have started using graph neural networks [44, 45, 46]. Various perspectives [47, 48] in trajectory prediction, such as knowledge distillation [49] and end-point prediction [6, 32], have also improved prediction performance. Trajectory imputation has not received much attention, with only a few studies addressing this task. NAOMI [10] is a non-autoregressive decoding process for deep generative models capable of imputing missing values for long-term spatiotemporal sequences. Furthermore, it uses a generative adversarial imitation learning objective. GMAT [50] is a hierarchical framework for sequential generative modeling that uses weak macro-intent labels. Both NAOMI and GMAT can handle the trajectory imputation task. In contrast, INAM [17] can handle both trajectory imputation and prediction tasks. It employs two models: one to predict future trajectories and supervise the other model, which learns to impute missing values in a non-autoregressive way. Both models are trained in an end-to-end fashion. Recently, GC-VRNN [8] proposed a unified framework that simultaneously imputes missing values and predicts future trajectories. It uses a Multi-Space Graph Neural Network with Temporal Decay to learn the temporal missing patterns. The datasets used by GC-VRNN and INAM for imputation are not publicly available.

### 2.3 Datasets for Pedestrian Trajectory Prediction.

With the advancement of pedestrian trajectory prediction research, various datasets have emerged (ref. Table 1). The most popular pedestrian datasets are ETH [20] and UCY [19], encompassing diverse behaviors such as social interaction, walking, and grouping. These datasets model pedestrian interactions across various scenarios. ETH and HOTEL datasets collectively contain 750 pedestrian trajectories, while UNIV, ZARA1, and ZARA2 datasets encompass 786 pedestrian trajectories, all in 2D coordinates. Another widely-used dataset is SDD [51], which includes pedestrians, bikers, skaters, carts, cars, and buses, totaling 10,300 trajectories. TrajNet [52] is a synthetic dataset that combines elements from the above datasets to create an interaction-centric trajectory-based dataset. The PETS [54] dataset consists of multi-sensor sequences depicting complex crowd scenarios, with coordinates extracted from 7 fps video images. The Grand Central Station [53] dataset captures crowd trajectories from scenes lasting 33.20 minutes at 25 frames per second, with coordinate trajectory data derived from scene images. Waymo [60], KITTI [57], and Argoverse [56] are primarily utilized in autonomous driving research, as they involve interactions between heterogeneous agents (e.g., vehicles and pedestrians) in urban environments. The inD [55] dataset, captured by drones, includes 8,200 vehicles and 5,300 vulnerable road users from four locations, featuring classes such as cars, trucks, bicyclists, and pedestrians. The Ko-PER [58] dataset, recorded using laser scanners and videos, features trajectories of people and vehicles at urban intersections. TRAF [59] is a dense and heterogeneous traffic video dataset with cars, bikes, pedestrians, rickshaws, and other road agents. There are several datasets for pedestrian trajectory prediction, as discussed above, but all of them assume that all coordinates are present in the observed trajectories. Our TrajImpute dataset, however, includes missing coordinates in observed trajectories to simulate real-world scenarios and offers a unified framework for evaluating both imputation and trajectory prediction methods.

## 3 TrajImpute Dataset

**Problem Formulation.** For the data generation, we consider the following trajectory prediction formulation where we represent the past observed trajectory with time stamp $T_{\text{ob}}$ as $\mathbf{x}_p = \{(x_p^t, y_p^t) \mid t \in [1, \ldots, T_{\text{ob}}]\}$, where $(x_p^t, y_p^t)$ is the coordinates of the $p^{th}$ pedestrian. There could be $N$ number of pedestrians where $p \in N$. So the original past observed trajectories data contain $\mathbf{X}_{org} = [\mathbf{x}_1, \mathbf{x}_2, \mathbf{x}_3 \ldots \mathbf{x}_N]$ trajectories where each $\mathbf{x}_p \in \mathbf{X}_{org}$. We follow the standard eight observed time frame of 3.2 seconds time length and twelve prediction time frame of 4.8 seconds.

**Data Generation.** To simulate real-time scenarios with missing coordinates in past trajectories, we introduce missing coordinates into the original observed trajectories of the ETH, HOTEL, UNIV, ZARA1, and ZARA2 datasets. Due to uncertainty, the missing values can be randomly generated from any given timeframe. To incorporate this uncertainty, we randomly induce missing values in past trajectories. Moreover, the temporal missing pattern contains several variations; for instance, there could be scenarios where consecutive coordinates are missing from the observations, alternate coordinates could be missing, or other possibilities. We ensure that our data generation covers these kinds of patterns. We follow two protocols for generating missing coordinates: an 'easy' protocol, where we induce missing coordinates in at most half of the total observed time frames, and a 'hard' protocol, where we induce missing coordinates in more than half of the time frames.

**Easy Protocol.** In the easy protocol, from the total observations spanning 8 time frames, at most half of the frames may contain missing observations; i.e., there are missing values in at most 4 of the time frames. Thus, the number of missing coordinates could range from 0 to 4. The pedestrian's observed trajectory may contain 0, 1, 2, 3, or 4 missing coordinates. For each trajectory $\mathbf{x}$, we introduce missing values as follows: (1) First, we uniformly randomly select the number of missing values, $m \in \{0, 1, 2, 3, 4\}$. (2) We then randomly choose $m$ unique indices from the set of past observed time frames $\{1, 2, \ldots, T_{\text{ob}}\}$ with uniform probability. Lets denote these indices as $\{i_1, i_2, \ldots, i_m\}$. (3) We set the coordinates of $\mathbf{x}$ at the selected indices to NaN as shown in Equation 1.

$$x_{i_j} = \text{NaN}, \quad \forall j = 1, 2, \ldots, m \tag{1}$$

We also create a mask that indicates the location of the missing values, with the mask value set to 1 if the coordinate is missing (NaN), and 0 otherwise.

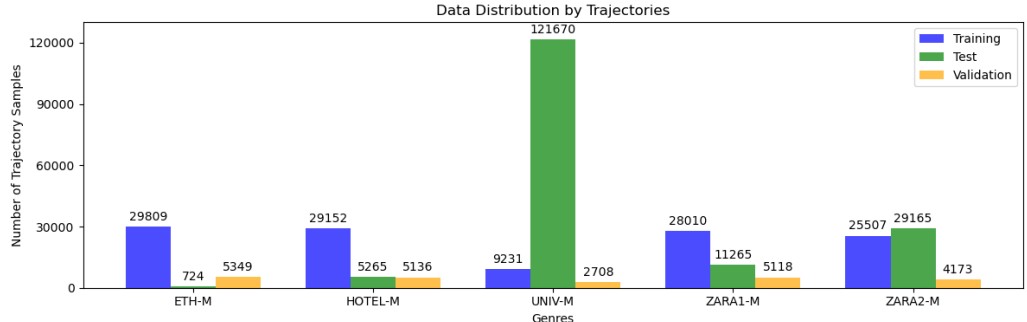

Figure 1: Number of trajectory samples in training, testing, and validation splits of the TrajImpute dataset.

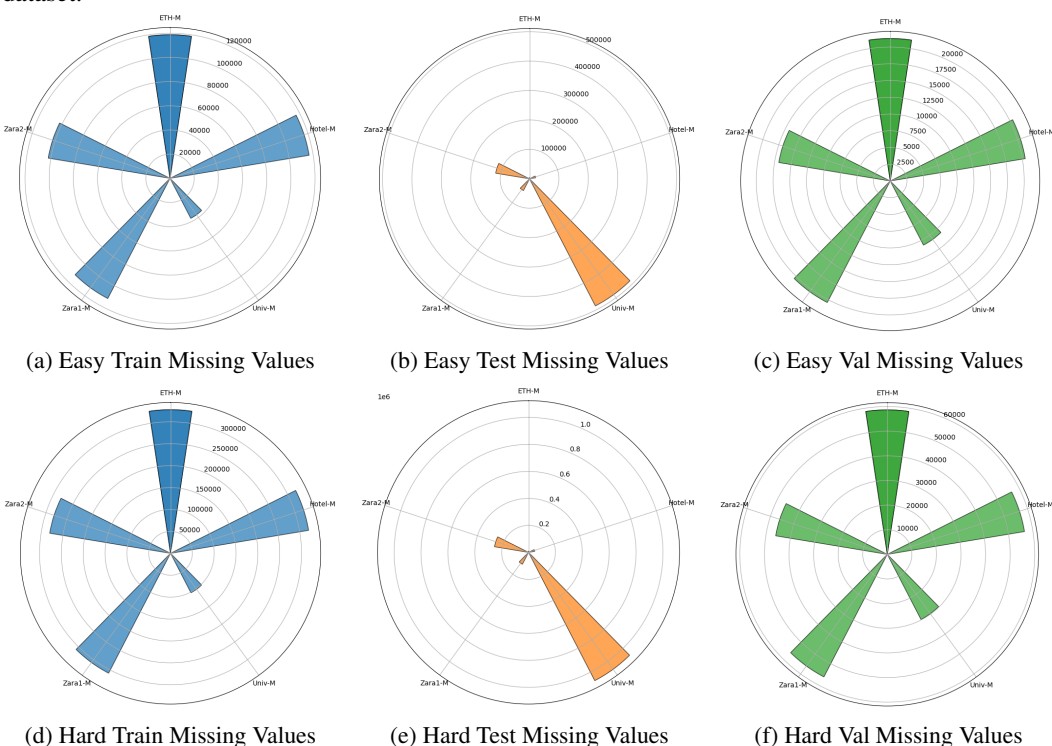

(a) Easy Train Missing Values    (b) Easy Test Missing Values    (c) Easy Val Missing Values

(d) Hard Train Missing Values    (e) Hard Test Missing Values    (f) Hard Val Missing Values

Figure 2: Illustration of the total missing coordinates in the easy and hard protocols for the ETH-M, HOTEL-M, UNIV-M, ZARA1-M, and ZARA2-M subsets of TrajImpute. 'M' refers to missing, indicating that the subset contains missing observed coordinates. The hard protocol creates more missing values compared to the easy protocol.

**Hard Protocol.** There could be scenarios where the trajectory prediction model is unable to acquire most of the observations. In such cases, the question of how the prediction is influenced becomes of interest to many. To simulate these scenarios, we choose to miss the majority of observed coordinates in the observed trajectories. Out of the total 8 observed coordinates, we generate trajectories with a minimum of 4 and a maximum of 7 missing coordinates. We induce 4, 5, 6, or 7 missing coordinates, thus $m \in \{4, 5, 6, 7\}$. For simulating the hard protocol, we follow the same strategy as shown in the easy protocol but increase the number of missing coordinates ($m$).

## 4 Data Analysis

**Training and Test Data.** Our aim is to simulate real-world scenarios while generating the TrajImpute dataset. In the training data, we maintain the original total number of trajectories as in the predefined training split of the ETH, HOTEL, UNIV, ZARA1, and ZARA2 datasets for both the easy and hard protocols. For instance, in the easy protocol, we generate training data by randomly dropping 0, 1, 2,

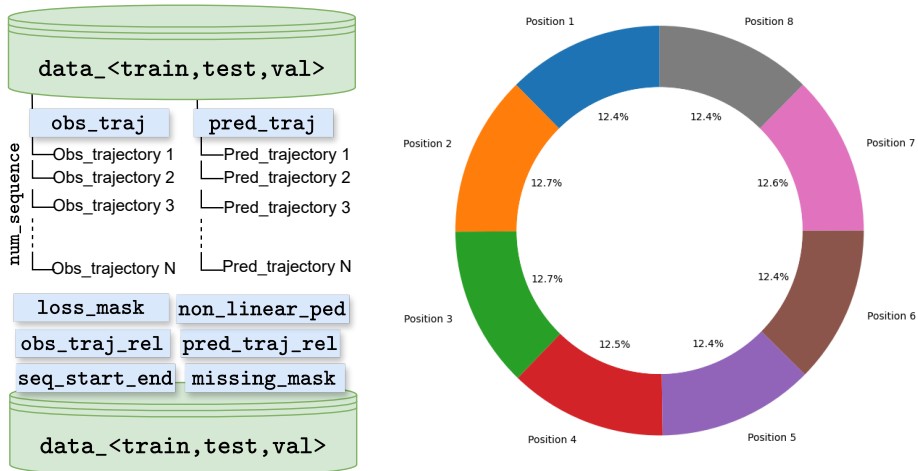

Figure 3: Illustration of an example (right) showing the missing pattern in the trajectory sequence of the ETH-M subset under the easy protocol. TrajImpute ensures that coordinates are equally likely to be dropped from the trajectory sequence, ensuring that any time frame can result in missing observations in the past trajectory. Furthermore, the structure of the TrajImpute dataset contains a dictionary structure with eight keys (left). Here, N is the number of trajectory sequences.

3, or 4 coordinates from each trajectory to create missing values. However, we increase the testing data to ensure fairness and consistency in testing. For instance, in the easy protocol, we generate test data as follows: first, we create $|m| = 5$ copies of each trajectory. The first copy contains no missing coordinates. In the second copy, we randomly drop 1 coordinate. In the third copy, we randomly drop 2 coordinates. In the same way, in the fifth copy, we randomly drop 4 coordinates. We repeat this process for all trajectories in the test data. Finally, we merge all these trajectories to obtain a test split for the imputation and trajectory prediction tasks.

**Data Statistics.** The total number of trajectories in the training, test, and validation splits in each subset of the TrajImpute dataset is shown in Fig. 1. Fig. 2 displays the total number of missing values in each subset of TrajImpute for both the easy and hard protocols. The hard protocol results in more missing values, whereas the easy protocol contains relatively fewer missing values. In TrajImpute, we ensure that the trajectory sequences capture the majority of the possible patterns/permutations of missing coordinates that could occur in the observed sequence. By sampling the missing indices from a random uniform distribution, each coordinate is equally likely to be missing, as shown in Fig. 3 (right).

**Structure of Data.** The TrajImpute dataset is built on top of the popular ETH, HOTEL, UNIV, ZARA1, and ZARA2 datasets, which collectively contain 1,536 human trajectories exhibiting diverse behaviors such as walking, crossing, grouping, and following. ETH and HOTEL include 750 pedestrians, while UNIV, ZARA1, and ZARA2 include 786 pedestrians. Following prior pedestrian trajectory prediction works [35, 32, 34], we use the same settings. The dataset structure follows the same format as previous works to ensure compatibility with existing pedestrian trajectory prediction methods. The observed trajectory length is 3.2 seconds, divided into 8 frames, and the predicted trajectory length is 4.8 seconds, corresponding to 12 frames. Missing observations can occur in the observed trajectory, so our missing values (NaNs) are only in the past observed trajectory with 8 time frames.

The structure of the TrajImpute dataset follows a dictionary format (Fig. 3 (left)) with specific keys, maintaining consistency across the training, testing, and validation splits. For instance, in `data_train`, there are 8 keys, each serving different purposes for the trajectory prediction task. The `obs_traj` key contains (values) the observed trajectories of the pedestrians, where each trajectory is a sequence of $(x, y)$ coordinates representing the pedestrian's position. We simulate missing/NaN values in the observed trajectories. The `pred_traj` key contains the predicted trajectories or ground truth trajectories, which do not contain missing values. The `obs_traj_rel` and `pred_traj_rel` keys represent the relative trajectories calculated by the finite difference/displacement from the previous to the current position in observed trajectory and predicted trajectories. The `missing_mask` key indicates where the missing values (NaNs) are present in the observed trajectory sequence.

Table 2: Results obtained for various imputation methods on the ETH-M, HOTEL-M, UNIV-M, ZARA1-M, and ZARA2-M subsets of TrajImpute with the easy protocol ($0 \leq$ missing $\leq 4$) and the hard protocol ($4 \leq$ missing $\leq 7$). The reported results show that SITES performs relatively better when imputing missing values. 'M' refers to missing, indicating that the subset contains missing observed coordinates.

| DATASETS | Methods | Metrics | Transformer [9] | US-GAN [62] | BRITS [22] | M-RNN [9] | TimesNet [16] | SAITS [14] |
|---|---|---|---|---|---|---|---|---|
| ETH-M | Easy-impute | MAE | 3.1318 | 0.6467 | 1.4287 | 5.2558 | 1.1353 | **0.5031** |
| | | MSE | 19.4576 | 1.8055 | 4.7339 | 35.3738 | 4.9441 | **0.9909** |
| | | RMSE | 4.4111 | 1.3437 | 2.1758 | 5.9476 | 2.2235 | **0.9954** |
| | | MRE | 0.5236 | 0.1081 | 0.2389 | 0.8787 | 0.1898 | **0.0841** |
| | Hard-impute | MAE | 3.2249 | 3.0451 | 3.0371 | 5.3309 | 1.3656 | **0.9965** |
| | | MSE | 19.5948 | 18.0716 | 17.9457 | 35.5047 | 4.9937 | **2.5934** |
| | | RMSE | 4.7926 | 4.2511 | 4.2362 | 5.9965 | 2.5054 | **1.6104** |
| | | MRE | 0.5734 | 0.5100 | 0.5087 | 0.8962 | 0.2287 | **0.1669** |
| HOTEL-M | Easy-impute | MAE | 8.8847 | 2.6327 | 3.9033 | 3.2133 | 7.4037 | **2.1930** |
| | | MSE | 91.5550 | 13.5993 | 23.1058 | 20.0857 | 124.5438 | **8.7460** |
| | | RMSE | 9.5684 | 3.6877 | 4.8068 | 4.4817 | 11.1599 | **2.9574** |
| | | MRE | 2.9468 | 0.8732 | 1.2946 | 1.0658 | 2.4556 | **0.7274** |
| | Hard-impute | MAE | 8.9096 | 7.8833 | 7.6057 | 3.2443 | 7.9484 | **2.6050** |
| | | MSE | 92.2607 | 75.9804 | 72.0169 | 20.2543 | 106.7010 | **16.0168** |
| | | RMSE | 9.6478 | 8.7167 | 8.4863 | 4.5005 | 11.3296 | **4.0021** |
| | | MRE | 2.8866 | 2.6127 | 2.5207 | 1.1686 | 2.6343 | **0.8634** |
| UNIV-M | Easy-impute | MAE | 3.0410 | 0.9158 | 1.0171 | 6.8380 | 0.6713 | **0.1939** |
| | | MSE | 14.0163 | 2.6297 | 2.9769 | 56.9715 | 0.7631 | **0.0697** |
| | | RMSE | 3.7438 | 1.6216 | 1.7254 | 7.5479 | 0.8736 | **0.2639** |
| | | MRE | 0.3905 | 0.1176 | 0.1306 | 0.8780 | 0.0862 | **0.0249** |
| | Hard-impute | MAE | 3.9795 | 1.9430 | 1.8028 | 6.9148 | 0.9421 | **0.6158** |
| | | MSE | 15.4244 | 6.1815 | 5.4057 | 57.6533 | 1.5827 | **0.6003** |
| | | RMSE | 3.9639 | 2.4863 | 2.3250 | 7.7268 | 1.2581 | **0.7748** |
| | | MRE | 1.0326 | 0.2495 | 0.2315 | 0.9751 | 0.1210 | **0.0791** |
| ZARA1-M | Easy-impute | MAE | 2.6288 | 0.4832 | 0.7307 | 5.1152 | 0.3125 | **0.2054** |
| | | MSE | 10.0109 | 0.8599 | 1.2306 | 34.9869 | 0.1768 | **0.0775** |
| | | RMSE | 3.1640 | 0.9273 | 1.1093 | 5.9150 | 0.4204 | **0.2784** |
| | | MRE | 0.4326 | 0.0795 | 0.1202 | 0.8417 | 0.0514 | **0.0338** |
| | Hard-impute | MAE | 2.7532 | 2.2846 | 2.3140 | 5.1921 | **0.5699** | 0.6277 |
| | | MSE | 10.1228 | 7.8216 | 8.0351 | 35.7821 | **0.6327** | 0.8287 |
| | | RMSE | 3.1816 | 2.7967 | 2.8346 | 5.9976 | **0.7955** | 0.9103 |
| | | MRE | 0.4463 | 0.3756 | 0.3805 | 0.8673 | **0.0937** | 0.1032 |
| ZARA2-M | Easy-impute | MAE | 2.1301 | 0.3861 | 0.5556 | 5.0905 | 0.2409 | **0.1314** |
| | | MSE | 7.3276 | 0.6212 | 0.8292 | 31.5674 | 0.1329 | **0.0385** |
| | | RMSE | 2.7070 | 0.7882 | 0.9106 | 5.6185 | 0.3645 | **0.1963** |
| | | MRE | 0.3524 | 0.0639 | 0.0919 | 0.8422 | 0.0399 | **0.0217** |
| | Hard-impute | MAE | 2.2840 | 1.8605 | 1.8051 | 5.1698 | 0.5031 | **0.3632** |
| | | MSE | 7.6342 | 5.8511 | 5.5953 | 32.3531 | 0.6525 | **0.4313** |
| | | RMSE | 2.8630 | 2.4189 | 2.3654 | 5.8994 | 0.8078 | **0.6567** |
| | | MRE | 0.3612 | 0.3077 | 0.2986 | 0.8786 | 0.0832 | **0.0601** |

`loss_mask`, `non_linear_ped`, and `seq_start_end` keys follow the format obtained from prior trajectory prediction data processing scripts [35, 32, 34, 61]. More details are given in the supplementary material.

## 5 Experiments and Benchmarking

In this section, we provide benchmarking of trajectory imputation and trajectory prediction methods. TrajImpute contains missing values in the observed trajectory sequences, and we experiment with several imputation models to fill in these missing values. We adopt various imputation methods (Sec. 5.1) for the task of trajectory imputation, i.e., filling in the missing coordinates. We then use the

Table 3: Results obtained for various trajectory prediction methods on the imputed subsets of TrajImpute. We report the ADE/FDE for the trajectory prediction task on the clean, soft imputed, and hard imputed protocols. 'Clean' refers to a subset with no missing coordinates. Performance degradation occurs when trajectory prediction is performed on the hard imputed subsets.

| Baselines | | GraphTern | LBEBM-ET | SGCN-ET | EQmotion | TUTR | GPGraph |
|---|---|---|---|---|---|---|---|
| ETH | Clean | 0.42/0.58 | 0.36/0.53 | 0.36/0.57 | 0.40/0.61 | 0.40/0.61 | 0.43/0.63 |
| | Easy-impute | 0.77/0.74 | 0.37/0.55 | 0.42/0.71 | 0.46/0.62 | 0.54/0.73 | 0.45/0.75 |
| | Hard-impute | 0.78/0.77 | 0.85/1.07 | 1.07/1.44 | 0.47/0.63 | 1.12/1.53 | 0.92/0.93 |
| Hotel | Clean | 0.14/0.23 | 0.12/0.19 | 0.13/0.21 | 0.12/0.18 | 0.11/0.18 | 0.18/0.30 |
| | Easy-impute | 0.15/0.25 | 0.13/0.20 | 0.14/0.23 | 0.65/0.68 | 1.31/1.66 | 0.19/0.31 |
| | Hard-impute | 1.68/1.42 | 3.31/4.13 | 3.21/3.92 | 0.72/0.74 | 3.36/3.95 | 1.89/1.70 |
| UNV | Clean | 0.26/0.45 | 0.24/0.43 | 0.24/0.43 | 0.23/0.43 | 0.23/0.42 | 0.24/0.42 |
| | Easy-impute | 0.27/0.47 | 0.30/0.51 | 0.29/0.51 | 0.37/0.61 | 0.31/0.49 | 0.25/0.44 |
| | Hard-impute | 0.50/0.51 | 0.64/1.01 | 0.77/1.21 | 0.39/0.70 | 0.59/0.85 | 0.53/0.50 |
| ZARA1 | Clean | 0.21/0.37 | 0.19/0.33 | 0.20/0.35 | 0.18/0.32 | 0.18/0.34 | 0.17/0.31 |
| | Easy-impute | 0.22/0.38 | 0.20/0.35 | 0.22/0.38 | 0.27/0.43 | 0.24/0.41 | 0.18/0.32 |
| | Hard-impute | 0.96/1.25 | 0.37/0.60 | 0.61/0.97 | 0.28/0.44 | 0.50/0.77 | 0.58/0.45 |
| ZARA2 | Clean | 0.17/0.29 | 0.14/0.24 | 0.15/0.26 | 0.13/0.23 | 0.13/0.25 | 0.15/0.29 |
| | Easy-impute | 0.18/0.30 | 0.16/0.27 | 0.17/0.29 | 0.36/0.54 | 0.25/0.37 | 0.29/0.30 |
| | Hard-impute | 0.37/0.44 | 0.27/0.43 | 0.41/0.63 | 0.37/0.55 | 0.33/0.50 | 0.36/0.34 |

imputed trajectories from the best imputation model to benchmark pedestrian trajectory prediction methods (Sec. 5.2).

## 5.1 Trajectory Imputataion

The performance of imputation methods is evaluated using four metrics: Mean Absolute Error (MAE), Root Mean Square Error (RMSE), Mean Square Error (MSE), and Mean Relative Error (MRE). It is important to note that only the values indicated by $mask$ in the inputs are used to calculate the imputation errors. For instance, $calc\_mae(prediction, target, mask)$ calculates the imputation error using the MAE metric. Here, prediction is the imputed trajectory, the target is the original trajectory, and the mask is the indicating mask, which indicates the imputed indices $\{i_1, i_2, \ldots, i_m\}$. More details on evaluation metrics are given in the appendix.

**Performance of Imputation Models.** We evaluate six imputation baselines for the task of trajectory imputation on the TrajImpute dataset. These baselines include SAITS [14], US-GAN [62], Transformer [23], TimesNet [16], BRITS [22], and M-RNN [9]. The results are reported in Table 2. We evaluate the imputation methods on both easy and hard protocols separately. On the ETH-M subset of the dataset, SAITS performs better than other methods. On HOTEL-M, SAITS achieves better imputation results on both the easy and hard protocols. On UNIV-M, SAITS performs much better on both protocols. On ZARA1-M, SAITS performs better on the easy protocol, while TimesNet performs better on the hard protocol. On ZARA2-M, SAITS outperforms the other imputation methods. In conclusion, SAITS performs relatively better than the other imputation methods, indicating that SAITS is able to reconstruct the missing coordinates with minimum errors most of the time among all compared methods for pedestrian trajectory.

## 5.2 Trajectory Prediction

We use the imputed data obtained from the best-performing imputation model on each subset of TrajImpute to perform the trajectory prediction task. We evaluate the performance of the trajectory prediction model using the Average Displacement Error (ADE) and Final Displacement Error (FDE) metrics. The ADE calculates the average Euclidean distance between the predicted trajectory and the

Table 4: Training Time (TT) and Test Inference Time (TIT) of various imputation methods on the ETH-M dataset. The training time is reported per epoch in minutes, and the inference time is reported in seconds.

| Transformer | US-GAN | BRITS | M-RNN | TimesNet | SAITS |
|---|---|---|---|---|---|
| TT: 2.14 min
TIT: 0.23 sec | TT: 5.72 min
TIT: 1.45 sec | TT: 2.93 min
TIT: 1.84 sec | TT: 2.61 min
TIT: 0.48 sec | TT: 1.15 min
TIT: 0.40 sec | TT: 0.22 min
TIT: 0.29 sec |

Table 5: Training and test time for trajectory prediction methods on the ETH-M dataset.

| Methods | Eqmotion | GraphTern | LBEBM-ET | SGCN-ET | TUTR |
|---|---|---|---|---|---|
| Test Time | 0.68 seconds | 4.10 seconds | 13.86 seconds | 5 seconds | 0.43 seconds |
| Training Time | 44.11 seconds | 55 seconds | 58 seconds | 62 seconds | 6.41 seconds |

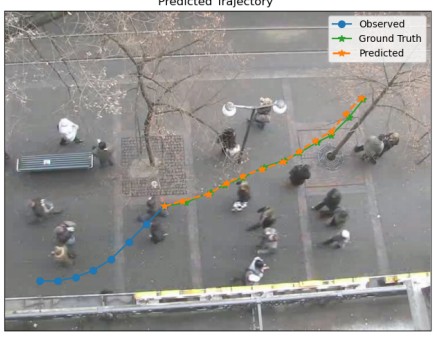

(a) Predictions from clean observations.

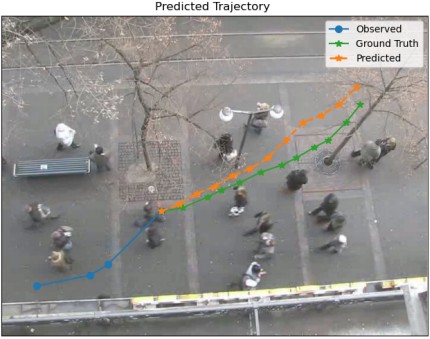

(b) Predictions from missing observations.

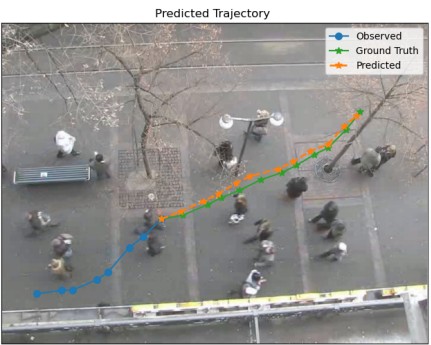

(c) Predictions from soft imputed observations.

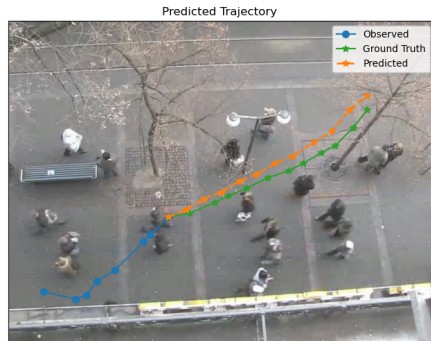

(d) Predictions from hard imputed observations.

Figure 4: Illustrations of predictions under different observation conditions: clean, missing, soft imputed, and hard imputed observations.

ground truth trajectory for each time frame, while the FDE calculates the Euclidean distance at the final time frame. More details are provided in the appendix.

**Performance of Prediction Models.** We evaluate recent trajectory prediction models on the imputed trajectory data, including GraphTern [6], SGCN-ET [61], EQMotion [63], TUTR [64], LBEBM-ET [61], and GPGraph [65]. The results are reported in Table 3. It is evident from the results that the trajectory prediction performance of all models tends to decrease with the *Hard-impute* setting. EQMotion has shown a comparatively smaller decline in performance when predictions are made on the hard imputed data. Training time and inference time of the imputation and trajectory prediction models are reported in Tables 4 and 5.

### 5.3 Qualitative Results

We visualize the trajectory predictions using the GraphTern model under four different settings: clean (no missing values), missing, Easy-Impute, and Hard-Impute (imputed using the SAITS method). Fig. 4b demonstrates that the poorest prediction results occur with missing observations. The prediction results in Fig. 4d improve compared to Fig. 4b when using hard-imputed observations. Additionally, the prediction results in Fig. 4c show a significant improvement over Fig. 4b when using soft-imputed observations, although they remain below the prediction results in Fig. 4a (clean observations – no missing values).

## 6 Discussion

From our comprehensive evaluation, we draw the following key insights: most of the imputation methods do not perform well on the task of imputing missing coordinates for pedestrian trajectories, except for SAITS. However, the performance of SAITS also starts degrading as the number of missing values increases (Hard vs. Easy). This indicates a need for imputation methods specifically tailored for pedestrian trajectory imputation. Furthermore, most of the trajectory prediction models do not perform well even in the Easy-impute setting, and all perform very poorly in the Hard-impute setting. Therefore, existing trajectory prediction models are unable to handle imputed data effectively. This indicates a need for developing trajectory prediction models specifically tailored either to directly handle missing coordinates present in observed trajectories or to perform better with imputed data. Thus, our TrajImpute dataset will foster future research in the trajectory prediction area.

**Broader Impact.** Our paper aims to bridge the gap between real-world scenarios and the rigid assumption that all coordinates are present in observed trajectories. By focusing on the challenge of anticipating and handling missing observed coordinates, we aim to enhance the effectiveness of trajectory prediction methods in real-world applications such as self-driving automobiles, robot navigation, human behaviour understanding, and more. A breakdown of the broader impact across different domains is given below:

1. For autonomous vehicles, trajectory prediction is essential for anticipating the movements of pedestrians and obstacles. Missing data due to sensor occlusions or failures can compromise the vehicle's ability to make safe decisions. Robust trajectory prediction under missing observations enhances safety by ensuring the vehicle can operate safely.

2. In dynamic environments, robots must predict the movements of humans, other robots, and objects to navigate safely. Missing data could lead to collisions or inefficient paths. Effective trajectory prediction under these circumstances ensures that robots can avoid obstacles and navigate more effectively.

3. In applications like crowd monitoring or event management, predicting the trajectory of people is vital. Missing data due to occlusions, sensor limitations, or other factors can lead to inaccurate predictions. Robust trajectory prediction can help manage large crowds more effectively, ensuring safety and efficiency.

**Limitation.** Future work can expand our work by incorporating new datasets, as mentioned in Table 1, and simulating the missing coordinates in these datasets, too. Imputation and trajectory prediction on these new datasets will provide additional insights to better access the performance of existing trajectory prediction methods.

## 7 Conclusion and Future Work

This paper introduces a trajectory prediction dataset with missing values in the observed coordinates. We conduct a comprehensive examination of several imputation methods to reconstruct these missing coordinates and benchmark their effectiveness for imputing pedestrian trajectories. Additionally, we analyze recent trajectory prediction methods and assess their performance on the imputed trajectories. Our experimental evaluation of both imputation and trajectory prediction methods yields several valuable insights. We empirically demonstrate the necessity of imputation methods specifically designed for pedestrian trajectory imputation. Moreover, our findings reveal that most trajectory prediction models perform poorly with imputed data, highlighting the need for developing models specifically tailored to handle missing coordinates for real-world applications.

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

# Appendix

## A Evaluation Metrics

**Mean Absolute Error**

$$\text{MAE (imputed, original, mask)} = \frac{\sum_{d=1}^{D} \sum_{t=1}^{T_{ob}} |(\text{imputed} - \text{original}) \odot \text{mask}|_t^d}{\sum_{d=1}^{D} \sum_{t=1}^{T_{ob}} \text{mask}_t^d} \tag{2}$$

**Root Mean Square Error**

$$\text{RMSE (imputed, original, mask)} = \sqrt{\frac{\sum_{d=1}^{D} \sum_{t=1}^{T_{ob}} \left(((\text{imputed} - \text{original}) \odot \text{mask})^2\right)_t^d}{\sum_{d=1}^{D} \sum_{t=1}^{T_{ob}} \text{mask}_t^d}} \tag{3}$$

**Mean Relative Error**

$$\text{MRE (imputed, original, mask)} = \frac{\sum_{d=1}^{D} \sum_{t=1}^{T_{ob}} |(\text{imputed} - \text{original}) \odot \text{mask}|_t^d}{\sum_{d=1}^{D} \sum_{t=1}^{T_{ob}} |\text{original} \odot \text{mask}|_t^d} \tag{4}$$

**Mean Square Error**

$$\text{MSE (imputed, original, mask)} = \frac{\sum_{d=1}^{D} \sum_{t=1}^{T_{ob}} \left(((\text{imputed} - \text{original}) \odot \text{mask})^2\right)_t^d}{\sum_{d=1}^{D} \sum_{t=1}^{T_{ob}} \text{mask}_t^d} \tag{5}$$

**Average Displacement Error** is calculated as the average of Euclidean distances between the predicted trajectory $\widehat{y}_i$ and the ground truth trajectory $y_i$ over all time steps $(m)$.

$$\text{ADE} = \frac{1}{m} \sum_{t=1}^{m} \left\| \widehat{\mathbf{y}_i}^{T_{n+t}} - \mathbf{y}_i^{T_{n+t}} \right\| \tag{6}$$

**Final Displacement Error** calculates the Euclidean distance at the final time frame $(T_{n+m})$.

$$\text{FDE} = \left\| \widehat{\mathbf{y}_i}^{T_{n+m}} - \mathbf{y}_i^{T_{n+m}} \right\| \tag{7}$$

