# Supplementary: Pedestrian Trajectory Prediction with Missing Data: Datasets, Imputation, and Benchmarking

**Pranav Singh Chib**[1]    **Pravendra Singh**[1]
[1]Department of Computer Science and Engineering
Indian Institute of Technology
Roorkee, India
{pranavs_chib,pravendra.singh}@cs.iitr.ac.in

## 1    Link for Datasets

Datasets and code files are publicly accessible at Link.

## 2    Hosting, Licensing, and Maintenance Plan

Our dataset will be hosted on both the GitHub and cloud storage drive. The dataset is provided under a Creative Commons CC BY-SA 4.0 license, allowing both academics and industry to use it.

## 3    Author statement

We (the authors) bear all responsibility including the violation of rights with respect to the release of this dataset.

## 4    Structure of Data

The structure of the TrajImpute dataset follows a dictionary format with 8 keys, including the `obs_traj`, `pred_traj`, `obs_traj_rel`, `pred_traj_rel` and `missing_mask`. We have already included these details in the main paper (see Section 4 and Figure 3 of the main paper). `loss_mask` is used to exclude some trajectories from contributing to the error calculation as used by prior works [1, 2]. `non_linear_ped` attribute refers to a flag/indicator that denotes whether the trajectory of a pedestrian (or any moving entity) is non-linear. `seq_start_end` denotes indices/pointers that mark the beginning and end of trajectory sequences.

## 5    Benchmarking Codes

### 5.1    Imputation Methods Codes

The following are the codes for the imputation methods used in our work.

- Code for the Transformer Link
- Code for the US-GAN Link
- Code for the BRITS Link
- Code for the M-RNN Link

38th Conference on Neural Information Processing Systems (NeurIPS 2024) Track on Datasets and Benchmarks.

- Code for the TimesNet Link
- Code for the SAITS Link

## 5.2 Trajectory Prediction Codes

The following are the codes for the trajectory prediction methods used in our work.

- Code for the GraphTern Link
- Code for the LBEBM-ET Link
- Code for the SGCN-ET Link
- Code for the EQmotion Link
- Code for the TUTR Link
- Code for the GPGraph Link

# 6 Datasheet

## 6.1 Motivation

The dataset is created to foster research on trajectory prediction with missing coordinates to provide realistic scenarios. The dataset is primarily created by an academic team (students and faculty).

## 6.2 Composition

The dataset contains instances of pedestrian trajectories represented in the form of coordinates moving along timestamps. The data statistics are shown in Section 4 of the main paper. The training and validation splits follow prior work in trajectory prediction, and the testing split is increased to assess the missing scenarios better (refer to section 3 of the main paper).

## 6.3 Collection Process

The data is generated using the existing ETH, HOTEL, UNIV, ZARA1, and ZARA2 pedestrian trajectory prediction datasets, and data generation scripts are given on GitHub at this Link. The framework used is Python 3.8.13 and PyTorch version 1.13.1+cu117. We use an NVIDIA RTX A5000 GPU with an AMD EPYC 7543 CPU.

## 6.4 Preprocessing/Cleaning/Labeling

Data preprocessing scripts are given on GitHub at this Link.

## 6.5 Uses

The dataset is intended for the pedestrian trajectory prediction task and related tasks with missing coordinates to provide realistic scenarios.

## 6.6 Distribution

The dataset will be made freely and publicly available and accessible. The dataset is licensed under a CC BY-SA 4.0 license.

## 6.7 Reproducibility

The baseline codes for both the imputation task and the trajectory prediction task are given on GitHub at this Link.