# OpenReview forum: "Pedestrian Trajectory Prediction with Missing Data: Datasets, Imputation, and Benchmarking"
_NeurIPS.cc/2024/Datasets_and_Benchmarks_Track — NeurIPS 2024 Track Datasets and Benchmarks Poster_

### Official Review · Reviewer_oGiB · 2024-07-12
**offical review**

**Rating:** 6
**Confidence:** 4

**Review:**

**Quality**

The research is well-executed with a strong emphasis on practical applicability. The experimental setup is robust, and the analysis is thorough, making the findings reliable.

**Clarity**

The paper is well-written and structured, making it easy to follow. However, the related work section could be more organized for better readability.

**Originality**

The introduction of TrajImpute is a significant step forward in addressing real-world challenges in trajectory prediction. However, the work mainly focuses on application and experimentation rather than introducing new theoretical models.

**Significance**

The dataset and the comprehensive evaluation provided can significantly impact the field by offering a realistic benchmark for future research.

1. While the paper introduces a new dataset and evaluation methods, it lacks theoretical innovation, mainly focusing on application and experimentation. There is a lack of systematic analysis of the principles behind trajectory prediction and imputation methods, and no new theoretical models are proposed. Most experiments and evaluations are based on existing methods, without introducing entirely new algorithms or models.
2. The third section of the related work lists many datasets but does not explain the specific characteristics and usage scenarios of each dataset in detail, nor does it provide a thorough comparison and contrast, resulting in insufficient analysis.
3. The structure of the related work section is unclear, with information piled together and lacking logical order and connecting words between works. Information could be organized by chronological order or usage categories to enhance clarity.
4. The dataset used in the experiments is relatively small, making it difficult to verify the effectiveness of the methods on larger-scale datasets.
5. Although various algorithms' performances are evaluated, there is a lack of in-depth analysis of their computational complexity and efficiency, such as model computational efficiency and robustness, to more comprehensively assess model performance.
6. The interpretation of experimental results is insufficient, lacking in-depth interpretation and analysis, for example, why certain models perform better than others on specific datasets.
7. The discussion on future work and improvements is minimal, without specific suggestions or research directions.
8. The transitions between paragraphs are not tight enough, lacking transitional sentences, making the paragraphs seem somewhat scattered.

**Strengths:**

1. The paper not only introduces the dataset but also provides a comprehensive evaluation of various imputation and trajectory prediction methods, offering a thorough experimental analysis.
2. By addressing the issue of missing data, the research is closer to real-world application scenarios, which is highly practical.
3. The dataset and code are publicly available, which helps to advance research in this field.
4. The experimental design includes both easy and hard missing data protocols, fully considering model performance under different levels of missing data, ensuring the reliability and comprehensiveness of the results.

**Additional Feedback:**

What are the specific methods and technical details used in the data generation process?

Why is it necessary to increase the test data?

Why does SAITS perform well in most cases, and what are its unique advantages?

**Clarity:**

The paper is generally well-written, though it could benefit from better organization in certain sections, particularly the related work.

**Correctness:**

The claims made in the submission are correct. The dataset is constructed in a sound way, and the evaluation methods and experimental design are appropriate and performed correctly.

**Documentation:**

The dataset documentation includes details on data collection, organization, availability, and intended use. Ethical considerations are also addressed.

**Ethics:**

There are no significant ethical concerns with this submission. The data used is publicly available, and the authors have ensured it conforms to ethical guidelines.

**Limitations:**

The authors have addressed some of the limitations in their analysis, but they could further improve by providing a more detailed analysis of computational complexity, data scale, theoretical foundations, and potential societal impacts.

**Opportunities For Improvement:**

1. Incorporating a more systematic analysis of the principles behind trajectory prediction and imputation methods could strengthen the paper.
2. Structuring the related work section more logically, perhaps chronologically or by categories, would enhance clarity.
3. Testing the methods on larger datasets to verify scalability and effectiveness.
4. Providing an in-depth analysis of the computational complexity and efficiency of the methods.
5. Offering more detailed interpretations of experimental results.
6. Providing a more detailed discussion on potential future work and research directions.
7. Improving transitions between paragraphs for better readability.

**Relation To Prior Work:**

The paper discusses how it differs from previous contributions, mainly by introducing a dataset that simulates missing coordinates, which is not commonly addressed in prior work.

**Summary And Contributions:**

This paper introduces TrajImpute, a dataset designed to simulate missing coordinates in pedestrian trajectory prediction. By introducing missing coordinates into existing pedestrian trajectory datasets and using various imputation methods to reconstruct the missing data, the authors evaluate these the performance of the methods on trajectory prediction tasks. The paper provides a comprehensive analysis of imputation methods and offers valuable insights into their effectiveness in trajectory prediction, thus bridging the gap between real-world scenarios and the assumption of complete observed trajectories.

---

> ### Author Rebuttal · Authors · 2024-08-17
>
> **General comment:** We would like to sincerely thank the reviewer for insightful comments and suggestions.
>
> >Significance 1: paper introduces a new dataset and evaluation methods, it lacks theoretical innovation
>
> **Response 4.1**: Please refer our Response 3.2.
>
> >Improvement 1: analysis of the principles behind trajectory prediction and imputation methods could strengthen the paper.
>
> **Response 4.2**: We apologize for this concern and appreciate the reviewer's suggestion. In response, we have provided additional analysis on the performance of imputation methods and trajectory prediction techniques. Please refer to our Responses 2.2 and 2.3.
>
> > Improvement 2: related work section
>
> **Response 4.3**:  We cover three different types of work in the related work section: trajectory prediction, imputation, and datasets. Due to space constraints, we are unable to include all the information. However, we have made significant efforts to provide important details such as the scenarios for dataset collection, information about the captured agents, and the modality. We have also included links to the datasets so that readers can find complete details. In response to your feedback, we will revise the related work section in the updated manuscript to improve readability.
>
> >Improvement 3: Datasets.
>
> **Response 4.4**: Please refer to our 'Common Response to Reviewers'.
>
> >Improvement 4 & 5: Computational complexity and efficiency / interpretations.
>
> **Response 4.5**: We apologize for the concern, and thank you for your suggestion. In response, we have now provided the computational efficiency in Tables 1 and 2 of our response (Please refer to our Response 2.4). Furthermore we have also provided the visual interpretation of the methods (Please refer to our Response 2.5).
>
> >Improvement 6 & 7: future work and research directions.
>
> **Response 4.6**: We apologize for the concern. As stated in the main paper, "Future work can expand our research by incorporating new datasets" and "there is a need for developing models specifically tailored to handle missing coordinates for real-world applications." We will provide more details in the revised version.
>
> We have included the additional limitations and additional future direction in responses 2.2, 2.3, and 2.7. Furthermore, we have also discussed the real-world applicability and impact of our work across different domains in response 1.1. For more details, please see our responses 1.1, 2.2, 2.3, and 2.7.
>
> >Feedback 1: data generation process
>
> **Response 4.7**: We apologize for any confusion caused. Our goal is to simulate real-world scenarios where missing values could be generated at any given time frame. As stated in Line 146 of the main paper, we aim to address all types of missing patterns in the observed coordinates, and we follow two protocols to ensure this. We will further elaborate on these points in the revised manuscript to enhance readability.
>
> >Feedback 2: Why is it necessary to increase the test data?
>
> **Response 4.8**: We apologize for any confusion. To ensure fairness and consistency in testing (refer to Line 175 of the main paper), we have increased the amount of test data for the sole purpose of covering all possible combinations. This is because there can be many different motion patterns, and missing coordinates can occur in various places with different frequencies.
>
> >Feedback 3: Why does SAITS perform well in most cases, and what are its unique advantages?
>
> **Response 4.9**:  We apologize for the confusion. We want to point out that, due to the attention mechanism, models that utilize attention often surpass RNN-based (BRITS, M-RNN) and CNN-based (TimesNet) approaches in imputation tasks [4], as they are better equipped to handle long-range dependencies. Furthermore, SAITS [5] incorporates dual joint learning tasks: a masked imputation task and an observed reconstruction task, which further capture both the temporal dependencies and feature correlations between trajectory time steps.
>
> [4] Wang, Jun, et al. "Deep learning for multivariate time series imputation: A survey." arXiv preprint arXiv:2402.04059 (2024).
>
> [5] Du, Wenjie, David Côté, and Yan Liu. "Saits: Self-attention-based imputation for time series." Expert Systems with Applications 219 (2023): 119619.

---

> > ### Comment · Reviewer_oGiB · 2024-08-17
> >
> > Thanks for your reply. Some concerns have been solved but there remains some.

---

> > > ### Author Rebuttal · Authors · 2024-08-17
> > >
> > > Dear Reviewer,
> > >
> > > Thank you once again for your valuable comments and suggestions, and for taking the time to read our rebuttal. We were in the process of finalizing the rebuttal and were still revising it to address your concerns at the time of your response. To the best of our knowledge, we have now addressed all of your comments, and we sincerely request you to kindly go through our responses. If you have any further comments, we are committed to addressing them. Thank you once again for your time.

---

> > > > ### Comment · Reviewer_jgDP · 2024-08-19
> > > >
> > > > Thank you for your continued diligence and commitment to improving your work. I think the authors answered all of my questions and I think it meet the standards.

---

> > > > > ### Author Response · Authors · 2024-08-24
> > > > > **Thank you for reading our rebuttal**
> > > > >
> > > > > Dear Reviewer  jgDP,
> > > > >
> > > > > Thank you for taking the time to review our rebuttal. If you find that all your concerns have been sufficiently answered/addressed, we kindly request your consideration in raising the score. Your support is greatly appreciated.
> > > > >
> > > > > Best,
> > > > >
> > > > > Authors

---

> > > ### Author Response · Authors · 2024-08-24
> > > **Seeking Further Feedback on Our Responses**
> > >
> > > Dear Reviewer oGiB,
> > >
> > > Thank you again for your valuable comments and suggestions, which are very helpful to us. We have posted responses to your concerns.
> > >
> > > We understand that this is quite a busy period, so we sincerely appreciate it if you could take some time to return further feedback on whether our responses resolve your concerns. If there are any other comments, we will try our best to address them.
> > >
> > > Best,
> > >
> > > Authors

---

### Official Review · Reviewer_o6nr · 2024-07-23
**The paper provides a useful dataset and benchmarking framework for pedestrian trajectory prediction with missing data, but lacks sufficient innovation to meet the standards of top-tier conferences like NIPS.**

**Rating:** 4
**Confidence:** 4
**Clarity:** yes

**Review:**

Pros:
1. The paper provides a thorough evaluation of existing imputation and prediction methods on a new dataset with missing data.
2. It addresses the real-world problem of data missingness in pedestrian trajectory prediction, which is crucial for applications like autonomous driving and robotics.
Cons:
1. The paper lacks significant methodological advancements or novel theoretical contributions, primarily focusing on extending existing datasets.
2. The creation of the TrajImpute dataset, while useful, is an incremental improvement over existing datasets and may not be sufficient for top-tier conference standards.
3. The analysis of results could be deeper, particularly in understanding the performance variations across different methods and scenarios.

**Strengths:**

The submission provides a practical dataset and thorough benchmarking framework for addressing the critical issue of missing data in pedestrian trajectory prediction, which is highly relevant to the fields of autonomous driving and robotics.

**Additional Feedback:**

-

**Correctness:**

The claims made in the submission are correct, and the dataset is constructed soundly; the evaluation methods and experiment design are appropriate and performed correctly, though they could benefit from deeper analysis and discussion.

**Documentation:**

yes

**Limitations:**

The authors have not adequately addressed the limitations and potential negative societal impacts of their work; they should consider discussing the real-world applicability of their methods, potential biases in imputation, and the ethical implications of deploying these models in critical applications.

**Opportunities For Improvement:**

The work's limitations lie in its incremental contribution with limited innovation, reliance on existing datasets, and insufficient depth in result analysis, which may not fully meet the high impact and novelty required for the conferences.

**Relation To Prior Work:**

The submission briefly mentions how it extends existing datasets by introducing missing data, but it does not clearly differentiate its methodological contributions from previous work.

**Summary And Contributions:**

The submission introduces TrajImpute, a dataset simulating missing coordinates in pedestrian trajectories, and benchmarks various imputation and prediction methods on this dataset. It aims to enhance real-world applicability by addressing data missingness.

---

> ### Author Rebuttal · Authors · 2024-08-17
>
> > Pros : The paper provides a thorough evaluation
>
> **Response 3.1:** We would like to sincerely thank the reviewer for their positive feedback on the evaluation, new dataset, and addressing real-world problems.
>
> >Cons: extending existing datasets, theoretical contributions.
>
> **Response 3.2:**  We apologize for the confusion. According to the scope of the *NeurIPS 2024 Datasets and Benchmarks Track*, the proposed dataset can be designed based on previously available data, and the scope for contribution is not restricted to the creation of entirely new datasets. As stated in the official guidelines: "*New datasets, or carefully and thoughtfully designed (collections of) datasets based on previously available data*" [[Link]](https://neurips.cc/Conferences/2024/CallForDatasetsBenchmarks). The novelty of our proposed work lies not only in the datasets but also in the benchmarking analysis of existing systems, which is explicitly welcomed in this track. The official guidelines mention: "*1. Benchmarks on new or existing datasets. 2. Systematic analyses of existing systems. 3. Carefully and thoughtfully designed (collections of) datasets based on previously available data*"
>
> Additionally, we are quoting answer of question 1 from FREQUENTLY ASKED QUESTIONS of this track: "*The reviewing procedures of the main conference are focused on algorithmic advances, analysis, and applications, while the reviewing in this track is equally stringent but designed to properly assess datasets and benchmarks.*" We have adhered to these guidelines and ensured that our work aligns with the scope and nature of the *NeurIPS 2024 Datasets and Benchmarks Track*.
>
> For your remaining concern, please see our "Common Response to Reviewers". We sincerely hope the reviewer finds our response satisfactory and will reconsider their decision.
>
> >analysis of results could be deeper
>
> **Response 3.3:** We apologize for this concern and appreciate the reviewer's suggestion. In response, we have provided additional analysis on the performance of imputation methods and trajectory prediction techniques. Please refer to our Responses 2.2 and 2.3.
>
> We have already presented the main insights regarding the performance of the imputation and trajectory prediction methods in Sections 5.1 and 5.2. We will further emphasize and provide more discussion to enhance readability.
>
>
> >Improvement : Opportunities For Improvement
>
> **Response 3.4:** Please refer Responses 3.2, 3.3.
>
> >Limitations: real-world applicability of their methods, potential biases, ethical implications.
>
> **Response 3.5:**  We regret any inconvenience caused. We have already addressed the limitations and impacts in the main paper (line 252). Our focus was not on proposing a new method, but rather on providing insights into the limitations of existing imputation and trajectory prediction methods from a real-world applicability perspective. Predicting pedestrian trajectories is crucial for applications such as self-driving cars, robot navigation, and human behavior understanding. However, in real-world scenarios, sensor failures, limited field of view, and occlusion can result in missing observations, significantly affecting the performance of these models.
>
> We have included the additional limitations and additional future direction in responses 2.2, 2.3, and 2.7. Furthermore, we have also discussed the real-world applicability and impact of our work across different domains in response 1.1. For more details, please see our responses 1.1, 2.2, 2.3, and 2.7.

---

> > ### Author Response · Authors · 2024-08-24
> > **Seeking Further Feedback on Our Responses**
> >
> > Dear Reviewer o6nr,
> >
> > Thank you again for your valuable comments and suggestions, which are very helpful to us. We have posted responses to your concerns.
> >
> > We understand that this is quite a busy period, so we sincerely appreciate it if you could take some time to return further feedback on whether our responses resolve your concerns. If there are any other comments, we will try our best to address them.
> >
> > Best,
> >
> > Authors

---

### Official Review · Reviewer_jgDP · 2024-08-01
**Comprehensive Evaluation of TrajImpute Dataset for Pedestrian Trajectory Prediction with Missing Data**

**Rating:** 7
**Confidence:** 4
**Clarity:** The paper is well-written and structu…

**Review:**

This paper addresses a gap in pedestrian trajectory prediction by introducing the TrajImpute dataset, which includes missing values to simulate real-world conditions more accurately. The authors provide a thorough examination of multiple imputation methods and evaluate their effectiveness in reconstructing missing coordinates. Furthermore, the paper benchmarks recent trajectory prediction models using the best-imputed data, offering insights into their performance with incomplete data.

Pros:
1. The introduction of TrajImpute fills a significant gap in current datasets by including missing values, which enhances the realism and applicability of trajectory prediction models.
2. The authors conduct a detailed evaluation of several imputation methods and trajectory prediction models, providing a clear understanding of their strengths and weaknesses.
3. The experimental results highlight the need for specialized imputation methods and trajectory prediction models that can effectively handle missing data, offering valuable directions for future research.

Cons:
1. It focuses primarily on a subset of existing datasets. Incorporating additional datasets could provide a broader perspective on the generalizability of the findings.
2. The performance of imputation methods degrades with an increasing number of missing values. The paper could benefit from exploring potential improvements or alternative approaches to address this issue.
3. The evaluation reveals that most trajectory prediction models struggle with imputed data, even under easy conditions. More detailed analysis and suggestions for improving these models would be beneficial.

**Strengths:**

1. The introduction of the TrajImpute dataset addresses a critical need for more realistic trajectory prediction datasets, which is highly relevant to the broader research community.
2. The research is methodologically sound, with a comprehensive evaluation of multiple imputation and trajectory prediction methods.
3. The work has significant potential to improve real-world applications such as autonomous driving and robotics, contributing to safer and more reliable systems.

**Additional Feedback:**

1. Provide more detailed analysis and insights into the performance of different imputation and prediction methods, particularly under challenging conditions.
2. Suggest potential future research directions, including the development of new imputation methods and trajectory prediction models specifically tailored for handling missing data.
3. Discuss the broader impact of this work on various applications, such as autonomous driving, robotics, and human behavior understanding, and how it can contribute to improving these systems.

**Correctness:**

The claims made in the submission are correct. The dataset is constructed soundly, and the evaluation methods and experiment design are appropriate and performed correctly. The comprehensive evaluation and benchmarking provide a clear understanding of the current state of the art and the need for further research.

**Documentation:**

The dataset documentation is sufficient, providing detailed information on data collection, organization, availability, and maintenance.

**Ethics:**

There are no significant ethical concerns with the submission.

**Limitations:**

The authors have adequately addressed the limitations and potential negative societal impact of their work. They clearly identify the performance degradation of imputation methods and trajectory prediction models under different conditions and suggest the need for specialized methods tailored for pedestrian trajectory imputation. Future work could expand by incorporating new datasets and further evaluating imputation and prediction methods in more varied real-world scenarios.

**Opportunities For Improvement:**

1. Can you please provide a detailed analysis of the model's computational complexity, particularly in comparison to other mainstream models?
2. Can you please enhance the discussion on the interpretability of the model by providing visualization tools or methods to help understand the decision-making process?
3. Conduct more detailed ablation studies to analyze the contribution of each component of the model to its overall performance.

**Relation To Prior Work:**

The paper clearly discusses how this work differs from previous contributions. It highlights the gap in existing datasets and the need for imputation-centric datasets for pedestrian trajectory prediction.

**Summary And Contributions:**

The paper introduces a dataset TrajImpute for pedestrian trajectory prediction that incorporates missing values in observed coordinates to better simulate real-world scenarios. The authors evaluate several imputation methods and benchmark their effectiveness for filling in missing pedestrian trajectory data. Also, the paper analyzes recent trajectory prediction methods and assesses their performance on the imputed trajectories, offering insights into the development of models specifically tailored for handling missing data in trajectory prediction tasks.

---

> ### Author Rebuttal · Authors · 2024-08-17
>
> **General comment:** We would like to sincerely thank the reviewer for his or her insightful comments and suggestions.
>
> >Cons 1: Additional datasets
>
> **Response 2.1:** Thank you for your thoughtful feedback and suggestions! Please refer to our response "Common Response to Reviewers".
>
> >Cons 2: The performance of imputation methods degrades with an increasing number of missing values.
>
> **Response 2.2:** Thank you for the reviewer's suggestion. The imputation methods work well, but when the number of missing values increases, the imputation model tends to suffer due to the loss of information. Only a few methods, such as SAITS, can handle high levels of missing values compared to others, but we still observe a decrease in performance.
> We aim to highlight this problem in exiting methods, both in imputation and trajectory prediction.
>
> SAITS performs relatively better compared to other models because it incorporates two joint learning tasks: a masked imputation task and an observed reconstruction task. These tasks help capture both the temporal dependencies and feature correlations between trajectory time steps. SAITS models use an attention mechanism [1], which enables them to handle long-range dependencies better than RNN-based (BRITS, M-RNN) and CNN-based (TimesNet) approaches in imputation tasks.
>
> The part of your concern has been addressed in our Response 2.7"Additional Future Directions".
>
> We will definitely include this discussion in the revised version to improve readability.
>
> >Cons 3:  Trajectory prediction models struggle with imputed data
>
> **Response 2.3:** Thank you for your valuable suggestion. We want to point out that the accumulation of errors [2,3] causes the trajectory prediction model to struggle. The imputation of missing values, which may not be entirely accurate, creates a disparity between the imputed and actual values. This discrepancy introduces errors in the predictions, which tend to propagate in future predictions due to the recursive nature of trajectory prediction.
>
>
> Additionally, many trajectory prediction models use coordinate information to calculate additional attributes such as velocity and acceleration. The absence or disparity of coordinates significantly impacts the calculation of these attributes/features, affecting trajectory prediction performance.
>
>
> Therefore, improving the imputation of missing values is crucial to enhancing the model's performance. We will definitely include this discussion in the revised version to improve readability.
>
> The part of your concern has been addressed in our Response 2.7 "Additional Future Directions".
>
> >Improvement 1: Model's computational complexity
>
> **Response 2.4:**  We apologize for this oversight. In response, we have conducted experiments on the computational complexity, and the results are reported in the Tables 1 and 2.
>
> **Table** 1: Training time (TT) and Test Inference time (TIT) of various imputation methods on the ETH-M dataset. The training time is reported per epoch in minutes, and the test inference time is reported in seconds.
> | Transformer | US-GAN             | BRITS              | M-RNN              | TimesNet          | SAITS             |
> |-------------|--------------------|--------------------|--------------------|-------------------|-------------------|
> | TT: 2.14 min | TT: 5.72 min       | TT: 2.93 min       | TT: 2.61 min       | TT: 1.15 min      | TT: 0.22 min      |
> | TIT: 0.23 sec | TIT: 1.45 sec       | TIT: 1.84 sec       | TIT: 0.48 sec       | TIT: 0.40 sec      | TIT: 0.29 sec      |
>
>
> **Table**2: Training (per epoch) and testing time (Test set) for trajectory prediction methods on the ETH-M dataset.
>
> | **Methods**  | **Test Time** | **Train Time**   |
> |--------------|---------------|------------------|
> | Eqmotion     | 0.68 seconds  | 44.11 seconds    |
> | GraphTern    | 4.10 seconds  | 55 seconds       |
> | LBEBM-ET     | 13.86 seconds | 58 seconds       |
> | SGCN-ET      | 5 seconds     | 62 seconds       |
> | TUTR         | 0.43 seconds  | 6.41 seconds     |
>
> >Improvement 2 & 3: Visualization and additional ablation studies
>
> **Response 2.5:**  We sincerely appreciate the reviewer's suggestion. In response, to enhance the interpretability of the decision-making process, we have included additional experiments in trajectory prediction (using GraphTern model) visualization under clean (no missing values), missing, Easy-Impute, and Hard-Impute settings (imputed using SAITS method). We have also provided a script to visualize future predictions under different settings. The visualizations can be accessed via [Attached PDF], and the script can be accessed on [GitHub](https://anonymous.4open.science/r/TrajImpute-B58E/README.md).
>
> The attached PDF includes four figures. Figure 1 shows the trajectory prediction results using clean observations (no missing values). Figure 2 depicts the predictions in the presence of missing observations. Figure 3 illustrates trajectory prediction results when using soft-imputed observations. Figure 4 illustrates trajectory prediction results when using hard-imputed observations.  It is evident from these figures that the poorest prediction results are obtained when there are missing observations (Figure 2). The prediction results in Figure 4 improve compared to Figure 2 when using hard-imputed observations. Furthermore, the prediction results in Figure 3 significantly improve compared to Figure 2 when using soft-imputed observations, although they are still lower than the prediction results in Figure 1 (clean observations - no missing values).
>
>
> We will definitely include this discussion in the revised version. Additionally, we will include a detailed discussion of the components of the imputation and trajectory methods in the supplementary material of the revised manuscript.

---

> > ### Author Rebuttal · Authors · 2024-08-17
> >
> > > Additional Feedback 1: Aanalysis and insights into the performance of imputation and prediction methods
> >
> > **Response 2.6 :** Please refer to our Response 2.2 and 2.3.
> >
> > > Additional Feedback 2: Additional Future Research Direction
> >
> > **Response 2.7 :** In the future, the issue of imputation methods performing poorly with an increasing number of missing values could be addressed by developing more sophisticated end-to-end methods where imputation is integrated directly with the downstream trajectory prediction task. This integrated approach could be more effective. Additionally, an alternate approach for future work could be to develop trajectory prediction models that are robust or aware of imputation, and perform better with imputed data.
> >
> > We will definitely include this discussion in the revised version to improve readability.
> >
> > > Additional Feedback 3: Broader impact of this work on various applications
> >
> > **Response 2.8 :**  Please refer to our Response 1.1.
> >
> >
> >
> >
> > ### References
> > [1] Wang, Jun, et al. "Deep learning for multivariate time series imputation: A survey." arXiv preprint arXiv:2402.04059 (2024).
> >
> > [2] Ivanovic, Boris, et al. "Propagating state uncertainty through trajectory forecasting." 2022 International Conference on Robotics and Automation (ICRA). IEEE, 2022.
> >
> > [3] Bae, I. and Jeon, H.-G. A set of control points conditioned pedestrian trajectory prediction. In Proceedings of the AAAI Conference on Artificial Intelligence, volume 37, pp6155–6165, 2023

---

> > > ### Author Response · Authors · 2024-08-24
> > > **Thank you for reading our rebuttal**
> > >
> > > Dear Reviewer jgDP,
> > >
> > > Thank you for taking the time to review our rebuttal. If you find that all your concerns have been sufficiently answered/addressed, we kindly request your consideration in raising the score. Your support is greatly appreciated.
> > >
> > > Best,
> > >
> > > Authors

---

### Official Review · Reviewer_ZnSu · 2024-08-02
**#872 Review**

**Rating:** 8
**Confidence:** 4
**Correctness:** They appear so.
**Clarity:** The paper is very well written.

**Review:**

Overall this is a well written paper. It is logically well structured and addresses an interesting problem.

**Strengths:**

This is a very interesting elaboration on the standard trajectory prediction problem by introducing missing values in observed coordinates.

**Additional Feedback:**

Please see above.

**Documentation:**

Yes.

**Ethics:**

No.

**Limitations:**

Yes.

**Opportunities For Improvement:**

The only limitation I can see is whether this is a bit of a niche improvement on the standard pedestrian trajectory problem.

**Relation To Prior Work:**

Yes.

**Summary And Contributions:**

This paper introduces a new data set for the pedestrian trajectory prediction problem. This is a well studied problem. However, this data set focusses on "missing values in observed coordinates" This is an interesting elaboration on the trajectory prediction problem.

---

> ### Author Rebuttal · Authors · 2024-08-17
>
> **General comment:** We would like to sincerely thank the reviewer for his or her insightful comments and suggestions.
>
> >Improvement 1: niche improvement
>
> **Response 1.1:** Thank you for your thoughtful feedback and remarks. We believe our work offers broader applicability within the field of trajectory prediction. Our paper aims to bridge the gap between real-world scenarios and the rigid assumption that all coordinates are present in observed trajectories. By focusing on the challenge of anticipating and handling missing observed coordinates, we aim to enhance the effectiveness of trajectory prediction methods in real-world applications such as self-driving automobiles, robot navigation, human behavior understanding, and more. In real-world applications, future trajectory prediction depends on past observed trajectories. Our paper demonstrates that even when using the best imputation methods, there are still differences between imputed and actual observations. We have noticed a decrease in the performance of trajectory prediction models despite using the best imputation method. It is vital for the broader use of trajectory prediction systems in real-world situations that various autonomous systems operate safely.
>
>
> A breakdown of the broader impact across different domains is given below:
>
> ### 1. Self-Driving Automobiles
>
> **Safety:** For autonomous vehicles, trajectory prediction is essential for anticipating the movements of other vehicles, pedestrians, and obstacles. Missing data, due to sensor occlusions or failures, can compromise the vehicle's ability to make safe decisions. Robust trajectory prediction under missing observations enhances safety by ensuring that the vehicle can still operate safely.
>
> **Decision-Making:** Self-driving cars rely on accurate trajectory prediction to make decisions such as when to merge, stop, or change lanes. With missing observations, the ability to predict the trajectories of surrounding vehicles becomes critical for maintaining smooth traffic flow and avoiding accidents.
>
> **Reliability:** Improving trajectory prediction methods to handle missing data can lead to more reliable autonomous systems, increasing public trust and the wider adoption of self-driving technology.
>
> ### 2. Robot Navigation
> **Obstacle Avoidance:** In dynamic environments, robots must predict the movements of humans, other robots, and objects to navigate safely. Missing data could lead to collisions or inefficient paths. Effective trajectory prediction under these circumstances ensures that robots can avoid obstacles and navigate more effectively.
>
>  **Autonomous Exploration:** For robots exploring unknown environments (e.g., drones in search and rescue missions), missing data is common. Enhanced trajectory prediction enables these robots to continue operating effectively even when sensor data is incomplete or temporarily unavailable.
>
>
> ### 3. Human Behavior Understanding
>
> **Crowd Dynamics:**  In applications like crowd monitoring or event management, predicting the flow of people is vital. Missing data due to occlusions, sensor limitations, or other factors can lead to inaccurate predictions. Robust trajectory prediction can help manage large crowds more effectively, ensuring safety and efficiency.
>
> **Surveillance and Security:** In surveillance, predicting the behavior of individuals or vehicles can help in preemptively identifying suspicious activities. Missing observations, however, pose a challenge, and improving trajectory prediction under such conditions enhances the effectiveness of security measures.
>
> ###  4. Unmanned Aerial Vehicles (UAVs) and Drones
>
>  **Autonomous Flight:** Drones, especially in complex environments, rely heavily on trajectory prediction to avoid obstacles and navigate safely. Missing observations due to sensor limitations or environmental factors like fog or dust can be hazardous. Improved prediction models ensure safer and more reliable drone operations.
>
>  **Package Delivery:** For UAVs used in delivery services, accurately predicting trajectories even with missing data ensures timely and safe deliveries, avoiding collisions with buildings, trees, or other drones.

---

> > ### Author Response · Authors · 2024-08-24
> > **Seeking Further Feedback on Our Responses**
> >
> > Dear Reviewer ZnSu,
> >
> > Thank you again for your valuable comments and suggestions, which are very helpful to us. We have posted responses to your concerns.
> >
> > We understand that this is quite a busy period, so we sincerely appreciate it if you could take some time to return further feedback on whether our responses resolve your concerns. If there are any other comments, we will try our best to address them.
> >
> >
> > Best,
> >
> > Authors

---

### Author Response · Authors · 2024-08-17
**Common Response to Reviewers**

We sincerely thank the reviewers for reviewing our manuscript. We would like to outline some points regarding our work in response to the reviewers' concerns:

* We designed several protocols and settings that simulate real-world missing scenarios to construct the TrajImpute dataset. We deliberately chose the most cited and widely used pedestrian trajectory prediction datasets (Table 1 of the main paper) to build upon our TrajImpute dataset. This allows us to make one-to-one comparisons with recent popular published methods in top-tier conferences (CVPR, AAAI, ICCV, ECCV, others) over the same datasets. Our goal here is to highlight the severity of the problem identified in our work on trajectory prediction for real-world applicability and to show that recent methods are inadequate for handling missing coordinates even after imputing them.
* Furthermore, we extensively evaluated known best-performing imputation models and assessed their impact on trajectory prediction. We experimented with six imputation methods across five datasets using two different protocols, resulting in 60 different benchmarks. Additionally, we benchmarked six trajectory prediction methods under three different settings on five different datasets, contributing to 90 different benchmarks. The total benchmarks are 150.
* Since there was no existing work in the proposed settings, we had to create our datasets using previously available data. We then benchmark all best-performing imputation models from scratch. We also evaluate the trajectory prediction methods over imputed data from scratch. We invested considerable time in conducting these benchmarks and deriving insights and findings. In the future, this work can be extended to more datasets and methodologies to enhance the applicability of trajectory prediction in real-world scenarios.
* Additionally, to improve the interpretability of the decision-making process, additional experiments are included in trajectory prediction visualization under missing, Easy-Impute, and Hard-Impute settings. For more details, please refer to our Response 2.5.
* In order to compare the computational complexity of the imputation and trajectory prediction methods, we have assessed their training and inference times for direct comparison. Please see our Response 2.4 for further details.
* We provide additional analysis on the performance degradation of imputation methods and trajectory prediction models, along with a discussion on why they struggle with increased missing data. For more details, please refer to our Responses 2.2 and 2.3.
* Additional future research directions and the broader impact of our work on various applications are discussed in our Responses 2.7 and 1.1.

We will definitely include this discussion in the revised version.

---

> ### Comment · Reviewer_jgDP · 2024-08-19
>
> Thank you for your continued diligence and commitment to improving your work.

---

> > ### Author Response · Authors · 2024-08-24
> > **Thank you for reading our rebuttal**
> >
> > We thank you for reading our responses and supporting our paper.

---

### Author Response · Authors · 2024-08-17
**Positive Feedback from Reviewers**

We wish to express our heartfelt gratitude to the Area Chairs and Reviewers for generously investing their time and providing us with valuable feedback. We sincerely thank Reviewer $\color{red}ZnSu$, Reviewer $\color{red}jgDP$, Reviewer $\color{red}o6nr$, and Reviewer $\color{red}oGiB$ for their encouraging comments on our work. We are glad that our manuscript has been recognized with the following merits:

* **Dataset Contribution and Relevance:** As the reviewers highlighted, the paper addresses an interesting problem (Reviewers $\color{red}ZnSu$, $\color{red}jgDP$), fills a gap in pedestrian trajectory prediction (Reviewer $\color{red}jgDP$), and presents well-executed research (Reviewer $\color{red}oGiB$). Additionally, it offers valuable directions for future research (Reviewer $\color{red}jgDP$) and advances research in this field (Reviewer $\color{red}oGiB$).

* **Methodology:** The research is methodologically sound (Reviewer $\color{red}jgDP$), with a protocol design that ensures reliability (Reviewer $\color{red}oGiB$). Furthermore, the dataset is constructed soundly (Reviewers $\color{red}o6nr, oGiB$).

* **Writing and Organization:** Overall, the paper is well-written and well-structured (Reviewers$\color{red}ZnSu, jgDP, oGiB$), and easy to follow (Reviewer $\color{red}oGiB$).


* **Benchmarking and Evaluation:** Reviewers mentioned that the paper provides a comprehensive evaluation of multiple imputation and trajectory prediction methods (Reviewers $\color{red}jgDP, o6nr, oGiB$) and offers valuable insights (Reviewer $\color{red}oGiB$).

* **Real-world Applicability:** The work has significant potential to improve real-world applicability (Reviewers $\color{red}jgDP, o6nr$) and bridge the gap between real-world scenarios (Reviewer $\color{red}oGiB$).

---

### Decision · Program_Chairs · 2024-09-26

**Decision:**

Accept (Poster)

**Comment:**

This paper focuses on the pedestrian trajectory prediction task. To better simulate real-world scenarios such as occlusion, and sensor failure, the authors study an interesting problem: missing coordinates in the observed trajectory. And they utilize some protocols to generate missing values in observed coordinates on current existing pedestrian datasets to evaluate models. There are 3 positive scores (8, 7, 6) and 1 negative score (4) finally.
After reading the negative review of Reviewer o6nr and its rebuttal, AC agrees with the authors that this paper satisfies the scope of the NeurIPS Datasets and Benchmarks Track, i.e., 1), new datasets, or carefully and thoughtfully designed (collections of) datasets based on previously available data, and 2), the reviewing procedures of the main conference are focused on algorithmic advances, analysis, and applications, while the reviewing in this track is designed to properly assess datasets and benchmarks.
Nevertheless, AC recommends the authors revise their paper to address the potential confusion and provide more details and extensive experiments to make their benchmark more solid, e.g. taking into consideration the reviews of Reviewer oGiB and jgDP.